# Translational Implications for Radiosensitizing Strategies in Rhabdomyosarcoma

**DOI:** 10.3390/ijms232113281

**Published:** 2022-10-31

**Authors:** Silvia Pomella, Antonella Porrazzo, Matteo Cassandri, Simona Camero, Silvia Codenotti, Luisa Milazzo, Francesca Vulcano, Giovanni Barillari, Giovanni Cenci, Cinzia Marchese, Alessandro Fanzani, Francesca Megiorni, Rossella Rota, Francesco Marampon

**Affiliations:** 1Department of Oncohematology, Bambino Gesù Children’s Hospital, Istituto di Ricovero e Cura a Carattere Scientifico, 00146 Rome, Italy; 2Department of Clinical Sciences and Translational Medicine, University of Rome Tor Vergata, 00133 Rome, Italy; 3Units of Molecular Genetics of Complex Phenotypes, Bambino Gesù Children’s Hospital, Istituto di Ricovero e Cura a Carattere Scientifico, 00146 Rome, Italy; 4Department of Radiological Sciences, Oncology and Anatomical Pathology, Sapienza University of Rome, 00161 Rome, Italy; 5Department of Maternal, Infantile and Urological Sciences, Sapienza University of Rome, 00161 Rome, Italy; 6Department of Molecular and Translational Medicine, Division of Biotechnology, University of Brescia, 25123 Brescia, Italy; 7Department of Oncology and Molecular Medicine, Italian National Institute of Health, 00161 Rome, Italy; 8Department of Biology and Biotechnology “C. Darwin”, Sapienza University of Rome, 00185 Rome, Italy; 9Department of Experimental Medicine, Sapienza University of Rome, 00161 Rome, Italy

**Keywords:** rhabdomyosarcoma, radiotherapy, radiation therapy, radiosensitizers, radioresistance

## Abstract

Rhabdomyosarcoma (RMS) is the most common soft tissue sarcoma of childhood and adolescence that includes FP-RMS, harboring the fusion oncoprotein PAX3/7-FOXO1 and FN-RMS, often mutant in the RAS pathway. Risk stratifications of RMS patients determine different prognostic groups and related therapeutic treatment. Current multimodal therapeutic strategies involve surgery, chemotherapy (CHT) and radiotherapy (RT), but despite the deeper knowledge of response mechanisms underpinning CHT treatment and the technological improvements that characterize RT, local failures and recurrence frequently occur. This review sums up the RMS classification and the management of RMS patients, with special attention to RT treatment and possible radiosensitizing strategies for RMS tumors. Indeed, RMS radioresistance is a clinical problem and further studies aimed at dissecting radioresistant molecular mechanisms are needed to identify specific targets to hit, thus improving RT-induced cytotoxicity.

## 1. Introduction to Rhabdomyosarcoma

### 1.1. Histological, Molecular Classification and Related Clinical and Prognostic Features

Rhabdomyosarcoma (RMS) is a rare and aggressive mesenchymal-derived soft tissue sarcoma (STS), preferentially occurring in childhood and adolescence. Most cases occur before the tenth year of life, and are exceedingly rare, with a worst prognosis, in adults. Based on histological characteristics, the two major subtypes of RMS are the embryonal RMS (ERMS) and the alveolar RMS (ARMS); the pleomorphic RMS (PRMS) and the spindle cell/sclerosing RMS (SSRMS), which typically occur in adults and children, respectively, are two other very rare subtypes [1,2,3]. However, since the histological classification frequently leads to an ambiguous subclassification, a molecular-based analysis characterizing subtype-specific genetic aberrations have been applied, and RMS is classified as not expressing, “fusion negative” (FN-RMS), or as expressing the fusion protein PAX3-FOXO1 (P3F) or PAX7-FOXO1 (P7F), “fusion positive” (FP-RMS) [1].

FP-RMSs are more frequently ARMSs, characterized by the presence of chromosomal translocations t(2;13)(q35;q14) or t(1;13)(p36;q14), precisely generating fusion proteins [2] that, acting as transcriptional factors, aberrantly regulate the expression of several target genes including N-MYC, IGF2, MET, CXCR4, CNR1, TFAP2B and FGFR4 [2]. However, the expression of fusion-proteins it is not sufficient and other genetic alterations have been shown to be related to ARMS transformation, such as the amplification of the regions: (i) 12q13-15, including C/EBP-homolog, transcription factor CHOP/DDIT3/GADD153, sarcoma-amplified sequence, transmembrane 4 superfamily member SAS/TSPAN31, alpha 2-macroglobulin receptor A2MR/LRP1, Sonic hedgehog (SHH) pathway effector, zinc finger transcription factor GLI1, cyclin-dependent kinase cell cycle regulator CDK4 and p53 pathway modulator MDM2; (ii) 2p24 of chromosome 2, including N-MYC gene; (iii) 13q31-32 including GPC5 and C13ORF25 genes [2]. FP-RMSs are rarer than FN-RMSs, and can occur at all ages, preferring adolescents, and young adults with a median age of 6.8 to 9.0 years [3]. No differences between male and female, or geographic and racial distribution as well as other extrinsic factors have been described. FP-RMSs, more frequently occurring on extremities, followed by paraspinal, perineal and paranasal sites, are frequently metastatic at the diagnosis, prognostically worse than FN-RMS [4,5] in adults compared to younger patients, and if expressing the PAX7 fusion protein [6]. Notably, patients with FN-ARMS are clinically and molecularly indistinguishable from FN-RMS [7].

FN-RMSs, more frequently ERMS, express various driving mutations converging on a limited number of pathways, including RAS (HRAS, NRAS, KRAS), PIK3CA, NF1, FBXW7 and genes orchestrating the regulation of the cell cycle [8,9,10,11]. Furthermore, most FN-RMS shows loss of heterozygosity (LOH) at the 11p15 locus, the site of the IGF-2 gene [4]. However, 60% of ERMS remains with a completely unknown biology [12]. The prognosis of FN-RMS is mainly determined by the stage of the disease at the diagnosis, with patients between the ages of 1 and 9 years having a better prognosis compared to infants and adolescents, and adults having the absolute worst prognosis among the FN-RMS [13]. Interestingly, the pathways perturbed in FN-RMS are frequently aberrantly regulated in FP-RMS because of the ability of fusion proteins to activate several cell surface receptor tyrosine kinases (RTKs) upstream of these pathways [14], indicating some commonality in the molecular driving forces in RMS. Finally, SSRMSs are characterized by mutation on MYOD1, TFCP2, NCOA2, VGLL2, or CITED2 genes [15,16,17,18] whilst PRMSs are characterized by a complex karyotype and an absence of recurrent molecular alterations [4]. Notably, the classification of RMS is constantly evolving [19] and, more recently, novel subtypes of RMS have been described [17,20,21].

### 1.2. The Management of RMS Patients

The diagnosis of RMS generally follows the onset of variable signs and symptoms that mainly depend on the site of origin, the patient’s age, and the presence or absence of distant metastases. Thus, RMSs of the head and neck area are determined by a localized, painless, enlarging mass, that results in pain when regarding extremities, whilst RMSs of the bladder or prostate may present with hematuria and urinary obstruction [22]. The initial evaluation of patients with suspected RMS should include a standard blood test, imaging analysis of the primary tumor site through computer tomography (CT) or magnetic resonance imaging (MRI), a total body CT and/or [F-18]2-fluoro-2-deoxyglucose positron emission tomography (18F-FDG PET)/CT for systemic staging, bilateral bone marrow aspirate and biopsy of the lesion to conclude the diagnostic and staging processes [23,24]. Notably, the histopathological and molecular analysis should be performed by an experienced pediatric pathologist. Myogenic markers like desmin, skeletal alpha-actin, myosin, and myoglobin and early myogenesis transcription factors like MyoD and myogenin must be investigated [25]. Expression of fusion gene P3F or P7F is very useful to identify subsets of ARMS, and microarray genome-wide RNA expression can generate, through various statistical algorithms, “diagnostic signatures” of the FP- and FN-RMS categories [4]. The staging system globally utilized for the management of RMS is the TNM (tumor, nodes, metastasis) staging system and considers orbit, non-parameningeal head and neck, non-bladder and non-prostate genitourinary and biliary tracts as favorable sites (Table 1).

Different risk stratifications are currently used in United States and in Europe. In North America, the Children’s Oncology Group (COG) Soft-tissue Sarcoma Committee, based on TNM-related stage, post-surgical procedure clinical group and fusion status, distinguishes low-, intermediate-, and high-risk prognostic groups (Table 2) [26]. The European Paediatric Soft Tissue Sarcoma Group (EpSSG) RMS2005 study [27], identifies low, standard, high, and very high-risk prognostic groups (Table 3). The treatment is based on surgery that should be limited to patients with operable RMS and in which it is possible to guarantee organ preservation; aggressive surgery is no longer recommended [28]. Thus, surgery remains critical for paratesticular RMS [29] and staging of lymph nodes in patients with RMS of extremity origin, and for any fusion-positive tumor [27]. Therefore, the standard care of RMS is based on multiagent chemotherapy (CHT), followed by radiotherapy (RT) or concomitant CHT and RT. Additional CHT can be recommended depending on the prognostic group [30,31]. According to the North American approach, CHT is based on vincristine, actinomycin-D, and cyclophosphamide (VAC) [31]. Alternating VAC to vincristine and irinotecan (VI) for patients with an intermediate-risk disease has been shown to give similar results but less toxicity [32]. Similarly, the European approach provides the use of eight cycles of a lower intensity vincristine and actinomycin-D (VA) for a low-risk disease, four cycles of ifosfamide, vincristine, and actinomycin-D (IVA) followed by five cycles of VA for the standard risk disease, nine cycles of IVA are used for a high-risk disease [30]. RT plays a critical role in the management and local control of RMS [33,34] as also suggested by the fact that omitting RT, the most common protocol deviation, is related to an increased risk of local disease progression and death [35]. Differences on the indications to RT exist between COG and EpSSG. Following COG indications, RT should be delivered after four cycles (12 weeks) of CHT [26], never beyond the 24th week or omitted [36]. The exception is the clinical group 1, FN-RMS [37]. Indications, doses, and target volume definition are summarized in Table 2, clinical groups have been defined by Intergroup Rhabdomyosarcoma Study Group (IRS) I–IV [38].

In agreement with EpSSG [44] and contrary to what was previously provided, the European approach does not recommend omitting RT, particularly for localized high-risk and very high-risk RMS. Surgery and/or RT should be performed after three cycles of induction CHT (3rd week), and only in some cases before. Metastatic RMS should be treated after 6 cycles of CHT, on primary and metastatic sites. RT can be omitted in R0 low-risk localized fusion-negative RMS, localized FN-RMS of the vagina with complete remission after induction CHT, standard-risk RMS occurring in a favorable site, resulting R0 after resection [44]. Indications, doses, and target volume definition are summarized in Table 3, clinical groups have been defined by Intergroup Rhabdomyosarcoma Study Group (IRS) I-IV [38].

A total of 24 Gy (1.5 Gy/fraction) of whole abdominopelvic RT for patients with peritoneal dissemination and/or malignant ascites [45] and 15 Gy (1.5 Gy/fraction) of whole lung RT for patients with lung dissemination [46] are recommended. In case of adult patients with RMS, RT can be delivered with a total dose of 50 up to 70 Gy [47,48,49]. Finally, a report from the Children’s Oncology Group (COG) Soft Tissue Sarcoma Committee has shown that intermediate risk RMS patients treated with induction chemotherapy followed by delayed primary excision can then receive a reduced dose of RT equal to 36 Gy if the margin status was negative or 41.4 Gy, if the margin status was positive or if patient was not candidate to delayed surgery but achieves a complete response after chemotherapy [50]. Notably, despite the technological improvements that have characterized RT in the last decade, local failures frequently occur. Thus, different schedules of irradiation (IR) have been tested, supposing that the solution was in the fractionation of the total dose used. However, hyperfractionating the dose, the use of smaller doses per single fraction per a larger number of fractions, investigated in the randomized IRS-IV study, failed [51], disavowing the erroneous idea that RMS had a poor ability to repair damaged DNA. On the other hand, the use of a hypofractinated schedule, larger dose per single fraction per a smaller number of fractions, did not improve the efficacy of RT [52,53,54,55,56,57,58,59,60,61,62,63,64]. Thus, in order to overcome radioresistance, new strategies able to target and destroy the molecular mechanisms responsible for radioresistance need to be identified.

## 2. Radiosensitizing Targets in RMS

Nowadays, radioresistance of RMS is still a clinical problem for cancer patients and oncologists. In recent years, several investigations have focused on the characterization of the main pathways involved in RMS radioresistance. This kind of research is extremely relevant because targeting radioresistance pathways could be a therapeutic strategy to improve the cytotoxicity induced by RT. From these studies, epigenetic targets, transcription factors, oncogenic pathways and DNA damage repair mechanisms acting as radiosensitizers have been already identified (Figure 1).

### 2.1. Epigenetic Targets

Epigenetic alterations play a key role in onset and progression of several human tumors, including RMS, especially through the transcriptional repression of tumor suppressor genes and the sustained expression of core regulatory transcription factors [65,66,67]. RT can induce epigenetic remodeling, therefore, the modulation of players responsible for reading, writing, and erasing the epigenome may impact on the cancer cells’ radiosensitivity.

#### 2.1.1. DNA Methyltransferases

DNA methylation is an enzymatic reaction which results in the addition of a methyl group at the carbon 5 position of cytosine, in the context of the sequence 5′ cytosine-guanosine (CpG). DNA methylation is involved in the regulation of many tissue-specific genes [68,69], as well as several important cellular functions [70]. Methylation of DNA is catalyzed by the DNA methyltransferase (DNMT) enzymes 1, 3A and 3B. DNMT1 is required for the methylation maintenance by preferentially methylating the unmethylated strand of hemimethylated DNA during replication. Conversely, DNMT3A and DNMT3B are necessary for the establishment of de-novo methylation of both strands during development [71]. Overexpression of DNMTs has been described in several human tumor types, including RMS [72]. Particularly, DNMT3A and DNMT3B have been associated with increased resistance to IR in a model of FN-RMS. Interestingly, they impact on RMS radiosensitivity in two different manners. Indeed, it has been recently demonstrated that, DNMT3A silencing enhanced RT related effects by triggering the senescence process [73]. Notably, it has been demonstrated that DNMT3A knockdown increases p21 and p16 levels, which in turn induce cell cycle arrest and senescence. Camero et al. hypothesized that DNMT3A silencing triggers cellular senescence stimulating temporary p21-mediated cell cycle arrest followed by p16 activation, thus causing permanent cell cycle arrest.

On the other hand, DNMT3B depletion increases radiosensitivity inducing DNA damage and affecting DNA repair mechanisms. DNMT3B silencing alone can induce DNA double strand break [73] and this effect, combined with the block of DNA repair, could be responsible for the increased sensitivity of DNMT3B-silenced cells. Indeed, DNMT3B downregulation resulted in decreased levels of crucial actors of the DNA repair machinery, such as ATM, RAD51 and DNA-PKcs. Interestingly, RT can induce the activation of these factors promoting DNA repair and cell survival, but DNMT3B silencing is able to counteract this effect, restoring FN-RMS cell sensitivity to IR [73]. These data suggest that treatment of FN-RMS cells with DNA-methyltransferase inhibitor, such as 5′-azacitidine, could be a promising strategy to increase the sensitivity to RT in RMS patients (Figure 2).

#### 2.1.2. Histone Deacetylases

Histone acetylation is one of the most common epigenetic modifications in human cells and is generally associated with open chromatin conformation. Acetyltransferases (HATs) and histone deacetylases (HDACs) regulate histone acetylation level by transferring or removing an acetyl group from acetyl CoA to the lysine residue, respectively, inducing, or repressing gene transcription [74]. Particularly, HDACs have been found overexpressed is several tumor types, including RMS, playing a crucial role in cancer onset and progression [75]. For these reasons, in recent years, interest in the inhibition of their activity as an anti-cancer strategy has increased exponentially. However, although HDAC inhibitors clinically improved outcome in patients with hematological malignancies, they failed in solid tumors [76]. Interestingly, several HDAC inhibitors showed radiosensitizing effects in RMS cells in vitro and in vivo. Indeed, a pan-HDAC inhibitor, PXD-101 or Belinostat, showed strong effectiveness in the induction of p21 and Cyclin B1 levels followed by G2/M cell cycle arrest in both FN- (RD, 0.41 μM) and FP-RMS (RH30, 0.23 μM) subtype [77]. Furthermore, Belinostat treatment increased apoptotic cell death by the activation of Caspase 9 and 3. Belinostat, by inhibiting the RAS/MEK/ERK signaling, led to a downstream deregulation of c-MYC expression both at transcriptional and post-transcriptional levels. Further, proof of Belinostat effectiveness on RMS cells was given by the decreased amount of cancer stem cell (CSC) population, as demonstrated by the reduction of CD133, CXCR4, Nanog and OCT3/4 levels. In addition, Belinostat treatment increased intracellular levels of reactive oxygen species (ROS) and impaired the non-homologous end joining (NHEJ) and the homologous recombination (HR) pathways, both involved in DNA repair [77]. All these data suggested that Belinostat treatment could increase the efficacy of IR of FN- and FP-RMS cells. Indeed, they demonstrated that a combination of Belinostat pre-treatment (40 mg/kg/dose) followed by RT on xenografted tumors (total dose of 12 Gy, 2 Gy/fraction, delivered three times per week) promoted an important reduction in tumor volume and weight compared to RT alone [77]. The observed effects were due to the Belinostat-dependent reduction of CHK1 and CHK2, key regulators of the cell cycle checkpoint that elicit a delay in the cell cycle progression to permit DNA repair [78].

Interestingly, different effects were observed when RMS cells were treated with the Class-I HDACs specific inhibitor FK288 or Romidepsin, which reversibly and not efficiently controls tumor proliferation of RMS cell lines in vitro [79]. Indeed, Romidepsin treatment on FN- (RD, 1.4 nM) and FP-RMS (RH30, 0.6 nM) cells up-regulates the expression of several positive cell cycle regulators such as Cyclin A, B and D1 in both subtypes and c-MYC, in FN- or N-MYC in FP-RMS cells, suggesting that Romidepsin is unable to inhibit RMS cell growth. Interestingly, Romidepsin treatment increases intracellular ROS levels and induces DNA damage in RMS cells radiosensitizing the FP-RMS subtype [79]. Similar results were obtained with the Class-I and -IV HDAC inhibitor, MS-275 or Entinostat. Cassandri et al. showed that Entinostat treatment induces G1 cell cycle arrest followed by irreversible cell growth arrest in a model of FP-RMS (RH30) [80]. Furthermore, Entinostat down-regulates the expression of the cell cycle positive regulators Cyclin A, B and D1 and up-regulates the expression of the cell cycle negative regulators p21 and p27 in both RMS cell subtypes (RD FN-RMS 1 μM; RH30 FP-RMS 1.9 μM). In addition, Entinostat treatment decreased the activation of MEK/ERK pathway in RD and AKT pathway and N-MYC levels in RH30. Furthermore, it has been demonstrated that Entinostat induces non-apoptotic cell death. Interestingly, as already demonstrated for Romidepsin, Entinostat treatment too is able to radiosensitize FP-RMS subtype to RT [80]. Indeed, Entinostat counteracts the ability of FP-RMS cells to repair the RT-induced DNA damage and detoxify from ROS accumulation, induced by RT, as demonstrated by the decreased levels of activated ATM, the key regulator of HR pathway, and by decreased mRNA levels of NRF2 and its targets, SOD, CAT and GPx4 [80]. Altogether these data demonstrated that HDAC activity inhibition could be a promising strategy to overcome RMS intrinsic radioresistance (Figure 2).

#### 2.1.3. Bromo- and Extra-Terminal Domain Proteins

Bromo- and extra-terminal domain (BET) proteins are epigenetic readers that regulate gene expression by recognizing acetylated histone proteins and recruiting to chromatin through two bromodomains (BD1 and BD2). The ubiquitously expressed BRD2, BRD3, and BRD4 and the testis-restricted BRDT belong to this family. BET proteins mainly localize at super-enhancers, enhancers, and promoters of active genes and, behaving as scaffolds, recruit other proteins and participate to the transcription elongation with the Mediator complex [81]. BET proteins are key activators of oncogenic networks involved in several cancer pathogenesis among which RMS [67]. It has been reported that mice with reduced BRD4 levels have severe effects in multiple tissues after exposure to IR [82]. In 2020, OTX015, a BET inhibitor (BETi) that selectively binds to BD1 and BD2, was tested in two FP-RMS cell lines (RH4 and RH30), showing that OTX015 treatment (1 and 3 μM) affects cell viability, cell cycle progression, stem cell self-renewal and migration ability of FP-RMS cells [83] (Figure 2). Moreover, BET inhibition activated apoptosis through the downregulation of AKT signaling and the induction of DNA damage. In agreement, it has been reported that OTX015 treatment increases RMS cell sensitivity to RT. Combinatorial treatment in RH30 cells pre-treated with 1 μM OTX015 for 24 h (h), and then exposed to 4 Gy IR, resulted in decreased cell proliferation and reduced colony formation capacity, coupled with a strong arrest in the cell cycle progression at G2/M phase and permanent DNA damage [83]. These findings suggest that OTX015 could counteract the RT-resistance phenotype by regulating proteins involved in the DNA repair pathway, thus allowing the accumulation of unrepaired DSBs and leading to cell death.

### 2.2. Transcription Factors

Upon IR, different cellular sensors perceive the DNA damage and activate intracellular signaling cascade, thus determining survival or death of the hit cells [84]. IR dose and dose rate influence and enhance the gene expression of several genes involved in the triggered signal transduction pathways. Transcription factors (TFs), by binding to DNA enhancers and promoters of target genes, regulate their expression and thus, targeting TFs could affect the choice between resuming physiological cell function after DNA repair or moving toward senescence/apoptosis [85].

#### 2.2.1. SNAI2

SNAI2 is a zinc finger transcription factor belonging to the Snail family (SNAI1, SNAI2 and SNAI3). It orchestrates important developmental biological processes by regulating cell function and differentiation in different tissues [86]. Moreover, SNAI2 regulates epithelial-to-mesenchymal transition (EMT), a transcriptional program that contributes to tumor progression and metastasis [87,88]. Accordingly, SNAI2 overexpression has been detected in several tumors [89] including RMS, in which, by competing with the master myogenic regulator transcription factor MYOD, promotes growth and blocks myogenic differentiation [90]. Moreover, SNAI2 has been described as a regulator of IR response and sensitivity through the activation of a SNAI2-dependent transcriptional response to DNA damage [91]. In 2021, Wang et al. demonstrated that SNAI2 levels directly correlates with radiosensitivity in both FN- (RH18, JR1, RH36, RD, SMS-CTR) and FP-RMS (RH28, RH30, and RH41) cell lines [92]. In agreement, alteration of SNAI2 expression (knockdown or overexpression) impacts on RMS radiosensitivity both in vitro and in vivo. Stable SNAI2-knockdown RH30, RD and RH18 cells exposed to 20, 15 and 10 Gy, respectively, showed a decreased proliferative rate and colony formation ability compared to single treatments. Furthermore, SNAI2 depletion combined with IR increased apoptosis and arrested cell cycle progression at G2/M phase. In an in vivo setting, combining SNAI2 lentiviral knockdown and IR (total dose of 30 Gy, 2 Gy/fraction; delivered five times per week for 3 weeks) on murine RH18 and RH30 xenografts resulted in earlier relapsed tumors post IR (7 week) in control xenografts compared to SNAI2-knockdown ones [92]. Authors reported that SNAI2 protects RMS cells from IR, by directly repressing the expression of the proapoptotic BIM, thus unveiling a p53-independent (nonfunctional in the cell lines used) SNAI2/BIM axis, potentially useful to predict IR treatment clinical responses and improve RMS therapy.

#### 2.2.2. C-MYC

MYC family members (c-MYC, N-MYC and L-MYC) are transcription factors that play crucial roles in several pathways needed for tumorigenesis (i.e., cell proliferation, differentiation, and genome stability [93]. Their overexpression has been associated with different human tumors [93] including RMS [94]. Increasing evidence reveals that c-MYC has a specific role in cancer stem cells and its epigenetic reprogramming increases the cancer stem cells phenotypes [95]. Moreover, c-MYC can impact on IR response by transcriptional activating CHEK1 and CHEK2, DNA-damage-checkpoint kinases, through a direct binding to their promoters [96]. In 2016, Gravina et al. tested the role of c-MYC in the transformed and radioresistant phenotype of FN-RMS cells [97]. In RD and TE671 cells, c-MYC expression or activity was hampered by a specific c-MYC shRNA, or a MadMyc chimera, respectively. C-MYC-targeted cells showed G1-phase cell cycle arrest, decreased growth rate, number and size of rhabdospheres, migration and invasion. Furthermore, they discovered that rapid (12 h) but not sustained (4 days) c-MYC targeting increased FN-RMS cells radiosensitivity (4 Gy total, delivered with a dose rate of 2 Gy/min) by promoting Caspase 3 and 9 IR-induced apoptosis. Increased γ-H2AX and decreased DNA-PKcs, RAD51 and Ku70 protein levels affected DNA damage repair in co-treated c-MYC- and IR-targeted cells [97]. Finally, the authors discovered that c-MYC directly interacted with the DNA-repair proteins RAD51 and DNA-PKcs and their silencing, with the latter as the most potent, radiosensitizes FN-RMS cells. Taken together these results indicate that c-MYC is involved in the radioresistant phenotype, and its targeting could ameliorate the therapeutic effects of IR (Figure 2).

#### 2.2.3. NRF2

Nuclear factor (erythroid-derived-2)-like 2 (NRF2) is a TF that regulates the expression of genes involved in cellular redox homeostasis and defense against oxidative stress [98]. In absence of oxidative stimuli, NRF2 is bound in the cytoplasm by Kelch-like ECH-associated protein 1 (KEAP1) and degraded in a proteasome-dependent manner. Upon oxidative stress, NRF2 is released from KEAP1 and can translocate into the nucleus where, by binding to antioxidant response elements (AREs) of its antioxidant target genes, activate their transcription [99]. In 2019 Marampon et al. reported that RD (FN-RMS) and RH30 (FP-RMS) cells treated with RT doses, ranging from 2 to 5 Gy, increased NRF2 transcript and protein expression [100]. The silencing of NRF2 combined with a single dose of 2 Gy IR resulted in a sustained presence of ROS levels 12 h after RT, compared to control-silenced cells. In accordance, NFR2-silenced cells showed downregulated levels of SOD-2, CAT and GPx4 genes, involved in the detoxification from ROS accumulation. Moreover, NFR2 silencing reduced colony formation ability and prevented RMS ability to restore γ-H2AX levels (biomarker for DNA-double strand breaks) 12 h after RT [100]. These findings suggest a role for NRF2 in activating the antioxidant program thus protecting RMS cells from ROS-induced DNA damage.

### 2.3. DNA Damage Effectors, Cell Cycle Regulators, and Cell Signaling Effectors

In general, the first cellular reaction to IR is the DNA Damage Response (DDR) activation. The DDR includes several signaling pathways that cause cell cycle arrest, DNA repair, and eventually apoptosis whether the lesions cannot be properly repaired. In cancer it is not uncommon to observe the overexpression of DDR proteins, as CHK1, as a countermeasure against the high rate of replication errors due to the increased division of tumor cells [101,102]. Thus, targeting proteins within DDR pathways could be critical to sensitize cancer cells to IR. Similarly, even targeting cell cycle regulators may be an excellent strategy to block cells in the phase of the cell cycle more susceptible to IR.

#### 2.3.1. PARP

Among the most studied DDR effectors, PARPs are included. Poly (ADP-ribose) polymerases (PARPs) are a family of enzymes involved in cellular differentiation, transcription, chromatin remodeling, DNA damage repair, cell death, and mitotic progression [103]. Particularly, PARP-1, -2, and -3 are involved in Single Strand Breaks (SSBs) and Double Strand Breaks (DSBs) repair, stalled replication forks, and DNA crosslinks. PARP-1 and PARP-2 act by the HR pathway, instead PARP-3 turn by NHEJ. PARP-1 recognizes and directly binds the DNA damaged changing conformation and increasing its catalytic activity. Moreover, PARP-1 forms poly-ADP-ribose polymers by adding ADP-ribose units to several proteins and altering their functionality [104]. The silencing of PARPs gene expression leads to synthetic lethality that could be used as therapeutic strategy to fight tumors harboring genetic mutations in DNA damage repair genes such as BRCA1, BRCA2, PTEN, and XCCR4. PARP-1 inhibitors (PARPi) increase the cytotoxic effects of IR reducing the survival and increasing the DNA damage of irradiated STS cells. In RMS A-204 cell lines, the treatment with PARPi Olaparib (1 μM), Iniparib (10 μM), or Veliparib (5 μM) combined with exposure to 2 Gy, 4 Gy, or 6 Gy of IR induces a synergistic effect on cellular survival reduction [105]. The combination of PARPi with X-rays induces a cell cycle arrest in the more sensitive G2-M phase of the cell cycle with respect to the two treatments alone. Furthermore, in RD (FN-RMS) and RH30 (FP-RMS) cell lines either the PARP 1/2 inhibitor, Olaparib, and the PARP 1/2/3 inhibitor, AZD2461, reduce cell proliferation. Even in this case, it is evident a consistent arrest in the G2/M as seen by morphological alterations and the increasing of volume cells. PARPi treatment determines the reduction of Cyclin D1, and the increase of apoptosis as indicated by the upregulation of p21 through the inhibition of AKT activation. Apoptosis is linked to DNA damage accumulation, highlighted by the persistence of γ-H2AX, the human marker of DNA damage, after 144 h of treatments. The cytotoxic effects induced by both PARPi treatments at high concentration (Olaparib 5 μM, and AZD2461 10 μM) occur mainly in RH30. Probably, the higher sensitivity to PARPi of RH30 is due to the lower PTEN levels or to the major levels of N-MYC with respect to RD cells. In RMS patients, PARPi renders cancer cells more sensitive to IR. Both AZD2461 and Olaparib increase the DNA damage induced by IR. Moreover, low concentrations of Olaparib (1.5 μM) and AZD2461 (5 μM) are more effective combined to IR in both RD and RH30 cells [106]. Taken together, these results indicate that the combination of PARPs inhibitors and RT can be explored in pediatric RMS clinical studies.

#### 2.3.2. Caveolin

Caveolin (CAV-1), a membrane-scaffold protein that contrasts the arrest in G2/M phase of the cell cycle and increases the senescence and apoptosis by reducing p21, p16, and the Caspase-3 cleavage [107], has first been correlated with poorly differentiated and a more aggressive phenotype in FN-RMS tumors [108,109,110]. Following the RT, CAV-1 acts as a radioprotective agent reducing DNA damage and increasing both ROS neutralization and DNA damage repair. Contrariwise, the loss of CAV-1 induces oxidative stress in the tumor environment through the increasing of H2O2 [111]. It has been demonstrated that human RD (FN-RMS) cell lines showing high expression levels of CAV-1 are resistant to RT. Moreover, in radioresistant cell lines as RDRR and RH30RR cells, high levels of phosphorylated CAV-1, probably mediated by Scr-kinases, have been observed [112]. Thus, the treatment with PP2 compound, a Src-kinase inhibitor, at the concentration of 20 μM combined to 4 Gy of IR sensitizes both radioresistant RDRR and RH30RR cell lines [113]. Indeed, CAV-1 can be considered a target that could be taken into account to upgrade the cytotoxic effects of RT.

#### 2.3.3. P53-MDM2 Pathway

p53 is a tumor suppressor that regulates different cellular pathways (i.e., DNA repair, cell cycle, apoptosis, senescence), playing a crucial role against cancer development/progression [114]. Conversely, the oncoprotein E3 ubiquitin ligase MDM2 negatively regulates p53 expression leading to its proteasomal degradation [115]. Either TP53 high mutational frequency and deletion, or MDM2 amplification leads to an impaired tumor suppressive role of the p53 pathway in cancer. Therefore, development of molecules aimed at hindering MDM2-p53 interaction is an attractive anticancer strategy. P53 mutations have been associated with defective IR response and radioresistance in different pediatric cell lines [116,117] and a recent study analyzed TP53 mutations and p53 pathway alterations (MDM2/4 amplifications and/or CDKN2A/B deletions) in 59 RMS patients that underwent RT to 126 sites [118]. Data demonstrate an association between TP53 mutations and increased irradiated tumor progression, with concomitant radioresistant phenotype and poor survival, suggesting a clinical relevance for p53 dysregulated pathway to improve RMS patient outcome.

In 2015 Phelps et al. tested the combinatorial effects of clinically relevant IR and the cis-imidazoline RG7112, an oral inhibitor of MDM2-p53 interaction, on CB17 SC female mice harboring either RH18 (FN-RMS, MDM2 amplified, TP53 wt) or RH30 (FP-RMS, TP53 wt) xenograft tumors [119]. IR treatment alone (2 Gy/fractions, 5-days per week) of 20 and 30 Gy in RH18 and RH30, respectively, induced regression with 100 percent tumor regrowth. Meanwhile, RG7112 oral gavage treatment alone (Schedule 1: 1 dose for 5 days; Schedule 2: 1 dose per week for three weeks) showed no antitumor activity in both the tested models. The combination treatment of IR and RG7112 (Schedule 1 and 2) hampered tumor regrowth in RH18 and enhanced time to recurrence in RH30, with no increase of IR-induced skin toxicity. Furthermore, combination treatments in FN- and FP-RMS xenograft models increased the expression of p53 downstream signals (such as p21, PUMA, DDB2) [119]. Altogether these data described the activity of RG7112 as an enhancer molecule of the daily-fractioned IR treatment in both FN- and FP-RMS models.

#### 2.3.4. MEK/ERK Pathway

The Ras/Raf/MEK/ERK pathway controls fundamental cellular signaling pathways (such as proliferation, differentiation, and survival) [120]. Mutation forms of any Ras pathway member have been detected in several human tumors [121] among which RMS (>50% of cases) [122]. Constitutively active Ras pathway has been involved in radioresistance through the activation of DNA-PKcs by the pro-survival PI3K-AKT pathway [123,124]. Moreover, the Ras pathway stabilizes the oncogenic transcription factor c-MYC [125] and in agreement, the Ras-mutated radioresistant phenotype is enhanced by c-MYC [126]. Marampon et al. in 2011 investigated the radiosensitizing effects of U0126, a MEK/ERK inhibitor, on FN-RMS RD, TE671 and the xenograft-derived RD-M1 cells [127]. FN-RMS cells were treated with a total dose of 4 Gy (delivered with a dose rate of 2 Gy/min) alone or in combination with 10 μM of U0126. The inhibition of MEK/ERK synergistically increases radiosensitivity by reducing the clonogenic survival and the protein level of Cyclin D1. Additionally, DNA-PKcs and c-MYC protein expression were affected by U0126 treatment alone or in combination with RT. In in vivo experiments, TE671 cells were xenografted in female CD1 athymic nude mice and treated with intraperitoneal injection of 25 mmol/kg U0126 (3 times per week, the day before RT) alone or in combination with a total dose of 12 Gy (2 Gy/fraction, delivered 3 times per week). The combinatorial treatment results in tumor mass reduction and delay in time of tumor progression, thus suggesting the synergistical effect of MEK inhibitor when combined with IR. In 2016, the same group tested the combinatorial efficacy of U0126 and IR on FN-RMS stem-like cell population [128]. U0126 treatment alone (10 and 40 μM), on RD and TE671 FN-RMS cells grown in stem-cell medium and anchorage-independent condition, reduced the stem-like phenotype (number and size of the rhabdospheres) and the associated markers (CD133, CXCR4 and Nanog). Combined treatment with U0126 (2 or 10 μM) and IR (total dose of 4 Gy, delivered with a dose rate of 2 Gy/min) further reduced the size and number of the spheres. Moreover, the expression of the stem-cell markers CD133 and CXCR4 were markedly reduced compared to single treatment, as well as the expression of the anti-apoptotic BMX. Altogether these results reveal that the cancer stem-cell potential of FN-RMS cells relays on ERK activation and, given the radioresistance phenotype might be the reason of cancer relapse, radiosensitivity induced by ERK inhibition could offer a promising therapy for FN-RMS patients (Figure 2).

#### 2.3.5. PI3K/Akt Pathway

The PI3K/Akt pathway, a key regulator of cell survival, is commonly associated with therapy resistance in cancer [129]. Dysregulation of Akt signaling is frequently observed in RMS [130], correlating with a poor overall survival [131,132]. Radioresistant RMS cell models have recently been shown to express higher levels of Akt1 phosphorylation on Ser473 in response to IR. Pharmacological inhibition of the PI3K/Akt signaling was indeed able to blunt the radioresistance [109]. Consistent with this, constitutive Akt1 activation in RD cells promoted a radioresistant phenotype by enhancing DNA repair through DNA-PK [133].

### 2.4. Genome Stability

Genomic instability is a hallmark of many types of cancer. It is characterized by an increased rate of genetic alterations including cytogenetic rearrangements, mutations, gene amplifications, and chromosomal aberrations. Moreover, IR exposure may enhance the mutation rate, facilitating the accumulation of the remaining genetic events required to produce a fully malignant tumor. Targeting components that guarantee genome stability as proteins involved in mitotic checkpoint, kinetochore-microtubule dynamics, and centrosome assembly could be a new approach to make RT more effective.

#### 2.4.1. KIF18B

Kinesin family member 18B is a protein associated with the pairing and separation of chromosomes during mitosis, the controlling microtubule length, and the centering of mitotic spindle [134]. In some sarcomas, KIF18b is increased after IR exposure facilitating their radioresistance to clinical therapies. Its high expression in sarcomas indicates a poor prognosis due to the activation of β-catenin [135]. By contrast, KIF18B-silenced sarcoma cells seem to be more sensitive to IR. In Sh-KIF18B transfected RD (FN-RMS) cells the cell survival fraction was significantly reduced with respect to sh-NC transfected controls after the exposure to 4, 6, and 8 Gy, while the apoptosis was increased only at 8 Gy [136]. A possible mechanism to explain the radiosensitivity of KIF18B-depleted RD cells could be the involvement of KIF18b in microtubule polymerization. It could act as Vincristine, a known microtubule destabilizer, that regulates the cell cycle. The reduction of KIF18B could induce the arrest of the cell cycle in the more radiosensitive phase G2/M. Moreover, similar results were obtained using the drug T0901317 [T09,N-(2,2,2-trifluoroethyl)-N-[4-[2,2,2-trifluoro-1-hydroxy-1-(trifluoromethyl)ethyl]phenyl]-benzenesulfonamide], the agonist of liver X receptor (LXR), indicating that the treatment with T09 combined to RT provides new insight into treatment of RMS.

#### 2.4.2. FANCD2

The human FANCD2 is a protein of Fanconi anemia complementation group (FANC). This gene displays a fundamental role into DNA repair pathways, especially those involved in repairing spontaneous DSBs. Mutations in FANC genes lead to Fanconi’s anemia (FA), a pathology characterized by increased cancer risk, bone marrow failure, and prenatal malformations [137]. Furthermore, FA patients are more sensitive to chemotherapeutic drugs and IR that create DNA inter-strand cross-links that cannot be accurately repaired [138]. Evidence from the analysis of more than 200 RMS patient tumor specimens indicated that the transcript and protein levels of FANCD2 are higher in FP-RMS tumors harboring the P3F fusion gene [139]. In vitro clonogenic survival assay demonstrated that the knockdown of FANCD2 in RH30 (FP-RMS) and RH18 (FN-RMS) provoked an increased sensitivity to IR [139]. Moreover, since FANCD2 is regulated by mTOR pathway [140], the inhibition of mTOR during RT has been considered as a radiosensitizing strategy. Accordingly, the combined treatment of AZD8055 (10 mg/kg via feeding), a potent mTOR kinase inhibitor [141], and RT (total dose of 20 Gy, 2 Gy/fraction) resulted in a significant improvement of survival in mice bearing RH30 but not RH18 xenograft [139]. Thus, the indirect targeting of FANCD2 by mTOR inhibition represents a novel approach for the treatment of FP-RMS tumors.

### 2.5. Cytokines and Receptors

Cytokines, polypeptides/glycoproteins with a low molecular weight (>30 kDa), mediate cell-to-cell communication involving growth, differentiation and pro- or anti-inflammatory signals. Cytokines bind to a corresponding set of receptors expressed by target cells, thus triggering intracellular signaling [142]. The Dose-dependent IR response can be modulated by cytokine-receptor pleiotropic effects (inflammation, invasiveness, fibrosis), thus becoming of particular interest to radiobiologist [143].

#### 2.5.1. HGF

The Hepatocyte Growth Factor (HGF) is a cytokine involved in liver generation as responsible of mature hepatocytes proliferation. HGF not only acts as a canonical grow factor but by binding to the c-MET tyrosine kinase receptor promotes the cell survival and tissues regeneration or suppresses the chronic inflammation and fibrosis [144]. In RMS, HGF is a chemoattractant that recruits cancer cells to bone marrow increasing their metastasizing ability. In particular, HGF increases the motility, polarity, adhesion, and cytoskeletal rearrangement of RMS cells and stimulates the Matrix metalloproteinases (MMPs) secretion. Indeed, targeting HGF could be pivotal to control the metastatic behavior of RMS cells. Moreover, the combination of HGF and RT or CHT increases the survival of metastasizing FP-RMS cells. The exposure of two c-MET-positive cell lines, RH30 and CW9019, to HGF (10 ng/mL) and 1500 cGy of γ-IR led to an increase in cell survival [145]. These results clearly indicate that the use of small-molecule inhibitors to suppress the c-MET-HGF axis represents a therapeutic approach not only to prevent the dissemination of RMS cells in bone marrow and lymph nodes but also to promote the success of RT.

#### 2.5.2. IFN-γ

In the 70s, angiogenesis inhibitors were considered to enhance the effects of chemotherapeutic agents as antiangiogenic drugs and cytotoxic agents could act synergistically on many different types of cancer [146]. The type-I interferons (IFNs) are pleiotropic cytokines with anticancer functions that exert as principal function the regulation of various immune cells. The antitumoral activity of type-I Interferons is known for a long time [147]. In particular, IFN-γ inhibits tumor growth and angiogenesis leading cancer cells to apoptosis. The angiogenesis inhibition depends on the normalization of tumor vasculature by reducing the vessel permeability and interstitial fluid pressure and improving tumor oxygenation. In vitro studies on RH30 and RH41 cell lines (FP-RMS) treated with recombinant human IFN-γ at concentrations of 30 to 3000 IU/mL for 24 h before IR at doses of 1, 2, or 4 Gy, showed that IFN-γ enhances the response of RMS to IR [148]. The improvement of tumor oxygenation induced by IFN-γ contributed to an increase in the effectiveness of IR, generating more oxidative stress. Moreover, IFN-γ seems to not affect the normal cell surrounding the tumor. In vivo experiments demonstrated that in RH30 FP-RMS xenograft the treatment with IFN-γ reduced the interstitial fluid pressure within the tumor to allow a better perfusion and oxygenation [148]. Thus, IFN-γ could be useful to expand the effectiveness of RT in clinical treatments of FP-RMS.

#### 2.5.3. Ephrin Receptor

The EPH (erythropoietin-producing hepatocellular) receptor tyrosine kinases (RTKs) and their ligands, the Ephrins, comprise a large class of signaling molecules involved in cell communications. The EPH/Ephrin cascade regulates several processes as cardiovascular and skeletal development, axon guidance and tissue patterning, myogenic differentiation of myoblasts and cell adhesion [148]. Alterations in the EPH/Ephrin network leads to several diseases such as cancer, indeed in several tumors including RMS, the EPH/Ephrin axis is upregulated [149]. Particularly, both FN- and FP-RMS globally overexpress EPH-B receptors and Ephrin-B ligands [150], while EPH-A receptors and Ephrin-A ligands are specifically upregulated in FN-RMS tumors and cell lines. Interestingly, EPH receptor inhibitors as GLPG1790, a small molecule that inhibits different EPH receptor kinases as EPH-A2 and EPH-B2, were addressed to arrest the tumor progression. This drug was tested in FN-RMS cells and showed radiosensitizing ability by affecting the DNA repair mechanisms induced by IR.

In vitro colony formation and wound healing assays demonstrated that RD (FN-RMS) treatment with 3.5 μM of GLPG1790 combined with 4 Gy of IR reduced cell migration and the ability to form colonies compared to control cells. Consistent results were obtained in an in vivo setting. Indeed, the administration of GLPG1790 (30 mg/kg) via gavage 5 days per week for 2 weeks sensitize mice, bearing RD xenografts, to RT treatment of a total dose of 12 Gy (2 Gy/fraction, delivered three times per week for 2 weeks) as demonstrated by the significant reduction of tumor weights and tumor progression with respect to controls [151]. Therefore, pharmacological treatments aimed at EPH/Ephrin inhibition combined to RT might be a therapeutic strategy to ameliorate FN-RMS tumor prognosis.

### 2.6. Other Sensitizing Agents

RT treatment mainly involves tumor sites, nonetheless normal tissues within the tumor or in its proximity receive clinically effective IR doses. One prophylactic approach is the administration of adjuvant agents, such as scavengers of ROS, aimed at reducing IR-mediated toxicity in normal tissues. Nonetheless, recent studies have demonstrated that ROS scavengers protect different normal cells form IR, while they can sensitize tumor cells by inducing cytotoxic effects [152].

#### 2.6.1. Selenium

Selenium is a free radical scavenger that showed potent antioxidant effect by reducing phospholipid hydroperoxides in the presence of the endogenous glutathione peroxidase [153]. Different studies have demonstrated selenium intake can reduce IR toxicity [152]. Moreover, topic or systemic administration of selenium combined with clinically relevant fractionated IR protocols, reduced IR-induced oral mucositis in a mouse preclinical model [154]. To date, the only in vivo study with fractionated IR protocols (total dose of 60 Gy, 2 Gy/fraction for 6 weeks), was performed in WAG/RijH rats subcutaneously isotransplanted with R1H RMS cells. Tumor growth delay, metastasis and repopulation rate were not affected by the intraperitoneal administration of selenium (15 µg/kg) 30 min before each IR [155]. The absence of selenium effects in combination with RT on RMS tumors could be ascribed to the intrinsic properties of the rat R1H RMS model known for decelerating and not accelerating the stem cell repopulation after RT [156]. Therefore, there is a need for more and well-detailed research studies to answer the question concerning the effects of selenium during fractionated RT.

#### 2.6.2. Resveratrol

Resveratrol (trans-3,4’,5-trihydroxystilbene, RES) is a natural chemical compound, synthesized by several types of plants in response to ultraviolet (UV) radiation exposure or mechanical stress induced by chemical or physical agents and pathogens. It is a polyphenol belonging to phytoalexins with several properties as promotion of anti-inflammatory response, antitumor activity, prevention of degenerative diseases, reduction of cardiovascular diseases, and inhibition of platelet aggregation. An excellent source of resveratrol is wine, especially red wine, or dark grape juice because of the high capacity of grape vines to produce resveratrol. The resveratrol exerts a protective effect at low concentrations and a sensitizing effect at high concentrations, indeed it is defined as a radio-modifying compound [157]. The protective effect is mainly due to the increasing of cellular detoxification from free radicals and ROS induced by the interaction of IR with living organisms. Generally, in human cancer cells resveratrol enhances cell proliferation (the number of cells) at low doses and decreases mitotic activity when used at high doses. In human RMS RD cells (FN-RMS), resveratrol exerts radioprotective effects at the concentration of 15 μM while it shows cytotoxic effects at 60 μM [158]. The combination of different concentrations of resveratrol (15, 30, and 60 μM) with doses of 50 Gy and 100 Gy of IR at different times post-IR (0 h, 24 h, 48 h), clearly shows that 15 μM resveratrol combined with 50 Gy of IR is capable to reduce the DNA damage after 48 h in RD cells and that the combination with 100 Gy renders the cells more resistant to DNA damage either after 24 h and 48 h. Interestingly, although Chow et al. demonstrated that doses of resveratrol between 50 μM and 100 μM reduced the cell proliferation of RD cells [159], the combination of 60 μM of resveratrol with high doses of IR does not show statistically significant cytotoxic effects [158]. Although resveratrol acts in a dose-dependent manner, further studies are needed to really understand whether the effect of RT on human RMS can be enhanced by the treatment with high concentrations of resveratrol or not.

#### 2.6.3. Fenretinide

Fenretinide (all-trans-N-(4-hydroxyphenyl) retinamide, 4-HPR) is a synthetic compound derivative of all-trans-retinoic acid used in clinical practice for the treatment of different tumors [160]. It acts as an atypical retinoid acid as it is capable to inhibit the cell growth by inducing apoptosis rather than differentiation [161]. In particular, in several cell lines fenretinide increased apoptosis and the production of ROS causing cell death in an independent manner from retinoic acid canonical behavior [162,163]. In RMS, although fenretinide treatment of RMS leads to apoptosis also in this case, these apoptotic processes depend on both ROS production and accumulation of cytoplasmic vesicles originating from macropinocytosis pathways [164,165]. The role of fenretinide as radiosensitizer for RMS treatment has been explored in a FP-RMS cell line. In RH4 cells (FP-RMS) the combination of 2 Gy IR and 0.5 μM fenretinide impaired the clonogenic growth capability. Significant reduction of cell viability in RH4 treated with 1.9 μM or 2.6 μM of fenretinide in combination with 5 Gy IR at 72 h, compared to fenretinide treatment alone, has been reported [164]. Moreover, the combinatorial treatment of IR and fenretinide, compared to single treatments, impaired cell cycle progression through G2/M phase, enhanced ROS production, and induced apoptosis. Specifically, increased apoptosis was due to the augment of macropinocytosis in RH4 cells treated with both IR and fenretinide as assessed by flow cytometry and light microscopy [164]. Thus, this evidence clearly indicates that fenretinide might be a promising agent that can be used in combination with RT for the treatment of FP-RMS cells.

## 3. Discussion

RT is currently a standard therapeutic strategy for RMS patients. Dose and dose rate in the clinical experience are modulated for both killing tumor cells, thus shrinking cancer mass, and reducing side effects on healthy tissues. Although innovative technologies greatly improve the IR delivery and reduce general toxicity, allowing complete remission in many RMS patients, recurrence may occur due to the insurgence of radioresistance mechanisms. Thus, there is still an urgent need to develop radiosensitizing strategies to ameliorate IR tumor response and consequently patient outcome.

Radiosensitizer agents (chemical or pharmaceutical) can improve the RT-killing effect by enhancing the induction of DNA damage and the production of ROS. Mostly, mechanisms of action involve the: (i) enhancement of DNA damage through the inhibition of the DNA repair pathways NHEJ and HR; (ii) impairment of cell cycle progression at a radiosensitive phase (G2/M); and (iii) gene expression alteration of IR resistance and IR sensitive genes. Further investigations are needed to better understand the radioresistance mechanisms and to develop new effective radiosensitizing strategies, aware that the degree of efficacy is dependent on the radiosensitizer type and dose, RT dose and timing and the interval of radiosensitizer and RT administration.

## Figures and Tables

**Figure 1 ijms-23-13281-f001:**
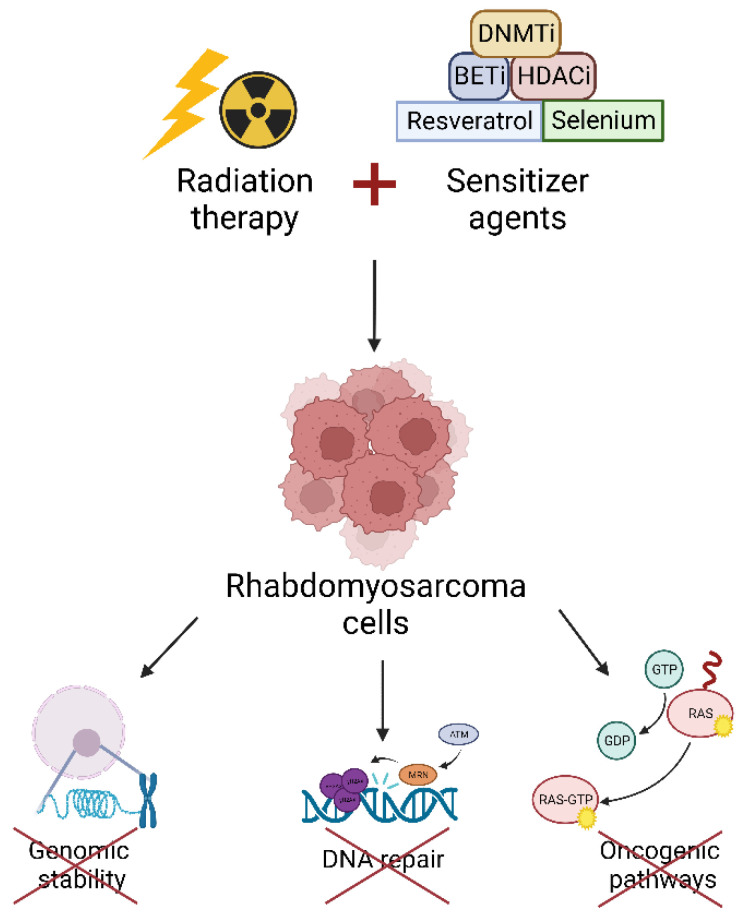
The combination of radiation therapy and sensitizer agents able to target epigenetic enzymes, transcription factors etc., sensitizes RMS cells to ionizing radiation by impairing genomic stability, DNA repair, and oncogenic pathways. DNMTi: DNA Methyltransferase inhibitors; BETi: Bromo- and extra-terminal domain inhibitors; HDACi: Histone Deacetylase inhibitors.

**Figure 2 ijms-23-13281-f002:**
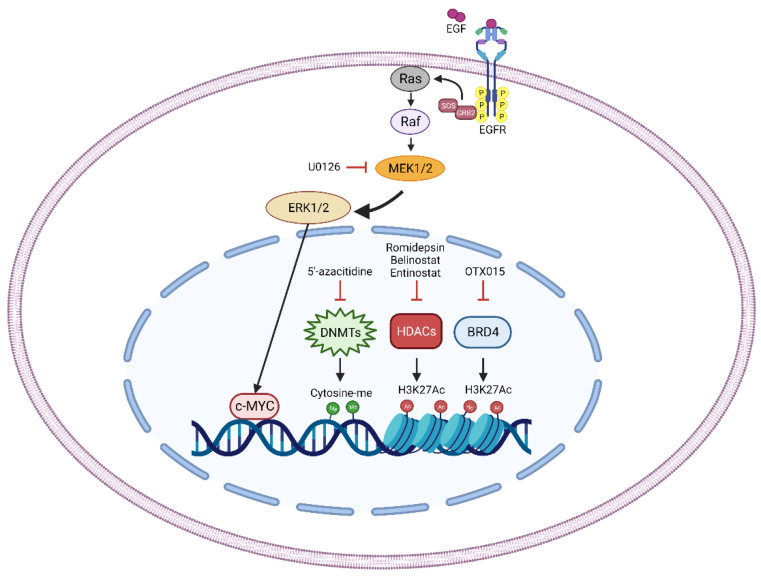
Schematic representation of the principal radiosensitizer targets (MEK/ERK pathway, DNMTs, HDACs, and BRD4) discovered in RMS. Treatment with U0126, a MEK inhibitor, decreases the activation of MEK/ERK pathway thus affecting the pro-tumorigenic abilities of the transcription factor c-MYC. The treatment with 5′-azacitidine reduces DNA methylation by specifically inhibiting DNA methyltransferases (DNMTs). Romidepsin, Belinostat, and Entinostat pharmacologically target Histone Deacetylases (HDACs) increasing the acetylation of Lysine 27 of the Histone 3 (H3K27ac). Bromodomain-containing protein 4 (BRD4) is directly inhibited by OTX015 treatment thus affecting its ability to read and bind the H3K27ac residues. Pharmacologic perturbation of the identified molecules increases RT-induced cytotoxic effects.

**Table 1 ijms-23-13281-t001:** TNM staging for RMS.

Stage	Sites	Tumor Stage		N	M
		Invasiveness	Size	N_0_ → No Nodes	
		T_1_ → Confined	a → < 5 cm.	N_1_ → Nodes Positive	
		T_2_ → Extended	b → >5 cm.	N_X_ → Unknown	
**1**	- Orbit	T_1_ or T_2_	a or b	Any N	M_0_
	- Head and neck: non-parameningeal				
	- Genitourinary: non-bladder non-prostate				
	- Biliary tract				
**2**	- Bladder/prostate	T_1_ or T_2_	a or b	N_0_ or N_X_	M_0_
	- Extremity				
	- Parameningeal				
	- Others: non-biliary tract				
**3**	- Bladder/prostate	T_1_ or T_2_	a b	N_1_ Any N	M_0_
	- Extremity				
	- Parameningeal				
	- Others non-biliary tract				
**4**	All	T_1_ or T_2_	a or b	N_0_ or N_1_	M_1_

**Table 2 ijms-23-13281-t002:** RMS prognostic groups based on Children’s Oncology Group (COG) soft-tissue Sarcoma committee.

Children’s Oncology Group (COG) Soft-Tissue Sarcoma Committee						
Risk Group	Stage (tnm)	Clinical Group (IRS)	FN-FP	Radiation Dose and Specific Indications		Ref.
		I → R0 + N0				
		II → R0 + N1 → R1 + N0 → R1 + N1				
		II → R2 → Only Biopsy				
		IV → Metastatic				
**Low—Subset 1**	1–2	I	FN	0 Gy		[37]
	1–2	II		36–41.4 Gy		[39]
	1	III Orbit		45 Gy	Complete Response after CHT	[40]
				50.4 Gy	Partial Response after CHT	
**Low—Subset 2**	1	III Non-Orbit	FN	50.4 Gy	≤5 cm	[41]
				59.4 Gy	>5 cm	
	3	I		0 Gy		[37]
	3	II		36–41.4 Gy		[39]
**Intermediate**	2–3	III	FN	50.4 Gy	≤5 cm	[41]
				59.4 Gy	>5 cm	
	4	IV ≤ 10 years old		Sites M^+^ and	See Risk Group	[42]
				15 Gy Whole Lung		
	1–3	I	FP	36 Gy		[43]
	1–3	II		36–41.4 Gy		[39]
	1–3	III		50.4 Gy	≤5 cm	[41]
				59.4 Gy	>5 cm	
**High**	4	IV	FN-FP	Sites M^+^	See Risk Group	[42]

**Table 3 ijms-23-13281-t003:** RMS prognostic groups based on European Pediatric Soft Tissue Sarcoma Group (EpSSG) RMS2005.

European Pediatric Soft Tissue Sarcoma Group (EpSSG) RMS2005						
Risk Group	Site	CLINICAL GROUP (IRS)	Nodal	FN-FP	Age/Size	Dose
		I → R0 + N0				
		II → R0 + N1 → R1 + N0 → R1 + N1				
		II → R2 → Only Biopsy				
		IV → Metastatic				
**Low**	Any	I	N0	FN	≤10 years	41.4 Gy
					≤5 cm	
**Standard**	Any	I	N0	FN	>10 years	41.4 Gy
					>5 cm	
	Favorable	II–III	N0	FN	Any	50.4 Gy
	Unfavorable	II–III	N0	FN	≤10 years	50.4 Gy
					≤5 cm	
**High**	Unfavorable	II–III	N0	FN	>10 years	50.4 Gy
					>5 cm	
	Any	II–III	N1	FN	Any	50.4 Gy
	Any	1-II-III	N0	FP	Any	50.4 Gy
**Very high**	Any	1-II-III	N1	FP	Any	50.4 Gy

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
