# Peer review of "Translational Implications for Radiosensitizing Strategies in Rhabdomyosarcoma"

_ijms, 2022, doi:10.3390/ijms232113281_

Round 1
Reviewer 1 Report
This is an interesting review on radiosensitizing strategies in rhabdomyosarcoma. The manuscript makes an exhaustive review on the different strategies to increase radiosesitivity of RMS cells. The review is exhaustive and very well written. In my opinion, the review is potentially interesting for readers given that it provides an integrated view of the effect of several pathways and proteins, as well as inhibitors, for radiosensitizing tumor cells.
Only some MINOR points need to be addressed:
MINOR POINTS TO BE ADDRESSED:
1- There are some (very few) minor spelling or orthographic errors that should be corrected.
- Line 116: “The treatment is based on surgery that should limited to patients […]”; it should read: “[…] should BE limited to patients […]”.
- Line 478: Please change “rabdospheres” to “rhabdospheres”.
- In my pdf version some (not all) greek letters (micromolar, beta-catenin..) are missing along the text. I guess this is a problem with the format since it is found many times along the text.
2 - In Figure 1, the Figure Legend should detail what are DNMTi, BETi and HDACi, since in this point of the text, it has not been explained yet.
3 – Line 385: Consider here, to not abbreviate “HR” pathway.
Author Response
We would like to thank Reviewer 1 for his/her accurate comments on our review.
We have addressed the indicated minor points as described below:
- Line 116: “The treatment is based on surgery that should limited to patients […]”; it should read: “[…] should BE limited to patients […]”.
We have changed accordingly the sentence as follow: The treatment is based on surgery that should BE limited to patients.
- Line 478: Please change “rabdospheres” to “rhabdospheres”.
We have changed “rabdospheres” to “rhabdospheres”.
- In my pdf version some (not all) greek letters (micromolar, beta-catenin..) are missing along the text. I guess this is a problem with the format since it is found many times along the text.
We apologies for the PDF formatting inconvenient. We went through all the manuscript and inserted all the missing greek letters.
- In Figure 1, the Figure Legend should detail what are DNMTi, BETi and HDACi, since in this point of the text, it has not been explained yet.
We would like to thank Reviewer 1 for pointed this out. We added the missing information in the Figure Legend 1 as follow: DNMTi: DNA Methyltransferase inhibitors; BETi: Bromo- and extra-terminal domain inhibitors; HDACi: Histone Deacetylase inhibitors.
– Line 385: Consider here, to not abbreviate “HR” pathway.
We would like to thank Reviewer 1 for this suggestion. We have already mentioned the extended Homologous Recombination and its HR abbreviation in the previous paragraph (see line 254). We then decided to keep HR abbreviation in line 385 (now line 402).
Reviewer 2 Report
This timely and very well written review by Pomella et al. provides a detailed overview of the current state of radiosensitization approaches for rhabdomyosarcoma.
First, the review introduces rhabdomyosarcoma classification very comprehensively, discussing the molecular features of the different subtypes, the different staging approaches followed in US and Europe, discussing the importance and limitations of radiotherapy for the management of the disease.
Next, the different targets important for radioresistance observed in patients are discussed. Epigenetic targets, on which the authors are experts, are discussed in particular detail with precision. Then, all the important known transcription factors are addressed in detail. Next, DNA damage, cell cycle, MEK/ERK and PI3K/Akt pathways are discussed in the context of RMS radiosensitization, with example of pre-clinical results with inhibitors.
Next, the authors discuss how genomic instability could be increased by targeting KIF18B and FANCD2, and how different cytokines and receptors (HGF, IFNg, EphRs) are involved and could be exploited for radiosensitization. Finally, they discuss other sensitizing agents (such as selenium, Resveratrol, and Fenretinide) that don’t fall into the above mentioned categories, but are nevertheless very important and promising.
In conclusion, this is a very timely, precise, and insightful review on radiotherapy and rhabdomyosarcoma, summarizing the state of the art, and proposing very interesting and novel paths to be investigated in order to increase the clinical success.
The figures are very clear and informative.
Major comment:
- Figure 2 legend needs to be expanded to better describe it, even if everything is already mentioned in the text.
- 106 Ref4 is given for diagnostic signatures, but this is a general review on RMS, please cite the papers describing the genetic signatures.
Minor comment:
- Fig2 definition and typesetting need to be revised, something must have happened during conversion to pdf.
- Greek symbols are missing throughout the text (conversion to pdf?)
- Typos:
o 85 NCOA2 is written twice
o 285 severe
o 370 comma
o 562 IFN
o 761 Ref 23 should probably be more complete
Author Response
We would like to thank the Reviewer 2 for his/her positive feedback on our manuscript.
Major cooment:
- Figure 2 legend needs to be expanded to better describe it, even if everything is already mentioned in the text.
We would like to thank Reviewer 2 for pointing this out and we agreed with this comment. We expanded the Figure Legend 2 as follow: Schematic representation of the principal radiosensitizer targets (MEK/ERK pathway, DNMTs, HDACs, and BRD4) discovered in RMS. Treatment with U0126, a MEK inhibitor, decreases the activation of MEK/ERK pathway thus affecting the pro-tumorigenic abilities of the transcription factor c-MYC. The treatment with 5’-azacitidine reduces DNA methylation by specifically inhibiting DNA methyltransferases (DNMTs). Romidepsin, Belinostat, and Entinostat pharmacologically target Histone Deacetylases (HDACs) increasing the acetylation of Lysine 27 of the Histone 3 (H3K27ac). Bromodomain-containing protein 4 (BRD4) is directly inhibited by OTX015 treatment thus affecting its ability to read and bind the H3K27ac residues. Pharmacologic perturbation of the identified molecules increases RT-induced cytotoxic effects.
- 106 Ref4 is given for diagnostic signatures, but this is a general review on RMS, please cite the papers describing the genetic signatures.
We have changed the Ref4 with the more appropriate citation:
- Davicioni, E.; Finckenstein, F.G.; Shahbazian, V.; Buckley, J.D.; Triche, T.J.; Anderson, M.J. Identification of a PAX-FKHR gene expression signature that defines molecular classes and determines the prognosis of alveolar rhabdomyosarcomas. Cancer Research 2006, 66, 6936-6946, doi:10.1158/0008-5472.CAN-05-4578.
Minor comment:
- Fig2 definition and typesetting need to be revised, something must have happened during conversion to pdf.
We would like to thank Reviewer 2 for this catch. We have replaced both the Figures with the high-resolution version.
- Greek symbols are missing throughout the text (conversion to pdf?)
We apologies for the PDF formatting inconvenient. We went through all the manuscript and inserted all the missing greek letters.
- Typos:
o 85 NCOA2 is written twice
We have deleted the NCOA2 repetition (now line 85)
o 285 severe
We have corrected “sever” with “severe” (now line 298)
o 370 comma
We have inserted comma in the paragraph title as follow: DNA damage effectors, Cell cycle regulators, and Cell Signaling Effectors.
o 562 IFN
We have corrected “INF” with “IFN” (now line 585)
o 761 Ref 23 should probably be more complete
We thank Reviewer 2 for pointing this out. We have completed the Ref 23 as follow: 23. Kaseb, H.; Kuhn, J.; Babiker, H.M. Rhabdomyosarcoma; In: StatPearls. Treasure Island (FL): StatPearls Publishing; 2022.